# Effects of Caffeine on Main Symptoms in Children with ADHD: A Systematic Review and Meta-Analysis of Randomized Trials

**DOI:** 10.3390/brainsci13091304

**Published:** 2023-09-11

**Authors:** Giuliana Perrotte, Marina Maria Gonzaga Moreira, Amauri de Vargas Junior, Alvaro Teixeira Filho, João Mauricio Castaldelli-Maia

**Affiliations:** 1Department of Neuroscience, Medical School, FMABC University Center, Santo André 09060-870, Brazil; jmcmaia2@gmail.com; 2Department of Medicine, Medical School, Centro Universitário Tiradentes, Maceió 57038-000, Brazil; moreiragmarina@gmail.com; 3Department of Medicine, School of Life Sciences, Pontifícia Universidade Católica do Paraná (PUCPR), Curitiba 82590-300, Brazil; avjunior91@hotmail.com; 4Department of Psychiatry and Behavioral Sciences, School of Medicine, University of Miami Miller, Miami, FL 33136, USA; alvaroteixeiramed@gmail.com; 5Department of Psychiatry, Medical School, University of São Paulo, São Paulo 05403-903, Brazil

**Keywords:** ADHD, caffeine, attention deficit hyperactivity disorder, neurodevelopmental disorders, children, pediatric

## Abstract

(1) Background: Attention-deficit/hyperactivity disorder (ADHD) is typically treated with stimulant medications, which may lead to several adverse effects. Recent animal studies have shown that caffeine can improve the symptoms of ADHD. This systematic review and meta-analysis sought to evaluate the effect of caffeine on ADHD symptoms in children. (2) Methods: PubMed, Embase, and Cochrane databases were searched for randomized controlled trials comparing caffeine with placebo in children, comparing overall symptoms of ADHD, inattention, hyperactivity, and impulsivity. (3) Results: We included seven RCTs in the systematic review for qualitative assessment, with 104 patients aged 5 to 15 years. Four of these studies (*n* = 76) were included in the meta-analysis. After qualitative analysis, four studies indicated no improvement in any of the ADHD symptoms compared with placebo. One study showed improvement in ADHD symptoms based on 1 of 5 scales applied. One study indicated significant improvement in general symptoms, inattention, and hyperactivity. One study indicated improvement in sustained attention but a worsening of impulsivity. In contrast, when using a quantitative analysis of the general symptoms of ADHD, the data showed no significant difference when comparing placebo with caffeine (standardized mean difference −0.12; 95% CI −0.44 to 0.20; *p* = 0.45; I^2^ = 0%). (4) Conclusion: overall, the totality of the evidence suggests no significant benefit of caffeine over placebo in the treatment of children with ADHD.

## 1. Introduction

Attention-deficit/hyperactivity disorder (ADHD) is a globally prevalent neurodevelopmental condition, affecting 5.3% of children [1], with boys being four times more affected than girls [2]. A global study across 204 countries demonstrated that ADHD ranks 84th in terms of disease burden represented by years lived with disability (YLD) across all age groups [3]. In childhood, the disorder is even more debilitating, ranking 39th among individuals aged 0 to 14 years [3].

The symptomatology of this disorder varies [4]. On the one hand, in children, the most common clinical presentation includes inattention, distractibility, hyperactivity, impulsivity, low academic performance, or behavioral problems at home or school [5]. On the other hand, adults with ADHD primarily experience difficulties in planning daily activities, extreme psychomotor restlessness, and harmful impulsivity [6].

The diagnosis can be made using the criteria outlined in the DSM-5 by the American Psychiatric Association [7]. In children and adolescents (under 17 years old), the disorder is identified when there are six or more symptoms present, in the inattentive and hyperactive-impulsive domains, or both [8]. In adults, a minimum of five symptoms in at least one of the three domains is required for diagnosis [9].

Patients with ADHD have catecholaminergic dysregulation, mainly in dopamine and noradrenaline functions [10]. The dopamine system in the cerebral cortex plays an important role in planning, novelty response, reward processing, and the initiation of motor responses [11]. Meanwhile, the noradrenergic system in the cerebral cortex influences the sleep-wake cycle, modulation of selective attention, inhibitory control, and cognitive preparation for urgent stimuli [12]. In individuals with this dysfunction, there is impulsive behavior and an inability to sustain attention [13].

Recent studies have shown that medication for ADHD treatment can reduce the risk of accidents and unintentional injuries in children, adolescents, and adults with ADHD [14]. Neglecting or not treating an ADHD person can result in academic impairment, low self-esteem, negative occupational outcomes, lower general adaptive functioning, peer rejection and aggression. Additionally, it can lead to risk behaviors and accidental injuries, such as traffic accidents, poisoning, fractures, traumatic brain injury, work accidents, and burns [15].

Pharmacological treatment is the first choice for children above 5 years old with ADHD [16,17,18,19]. First-line drugs include psychostimulants, while non-stimulants (such as guanfacine, atomoxetine, and clonidine) are considered second-line options for pharmacological treatment of ADHD in children above 5 years old [18]. Non-stimulants should only be considered if stimulants (methylphenidate and lisdexamfetamine) are not tolerated or do not provide symptom control benefits after two adequate trials [16].

Pharmacologically, methylphenidate acts on the dopamine and norepinephrine transporters, inhibiting the reuptake of these substances in the terminal neuron of the synaptic cleft and thus increasing their availability for synapses [20]. With more dopamine and norepinephrine present in the synapse, the transmission of these neurotransmitters is facilitated, improving communication between neurons [21]. This substance, however, can also cause decreased appetite, insomnia or sleep disturbance, nausea, tachycardia, increased blood pressure, mood swings, weight loss (at higher doses), and even possible growth reduction [22].

Lisdexamfetamine also increases the release of dopamine and norepinephrine in neuronal synapses [23]. However, this occurs through an action on the vesicular transport system, which is responsible for storing dopamine and norepinephrine in synaptic vesicles [24]. With greater release of these neurotransmitters, there is increased activation of their respective receptors, thereby helping to improve attention, concentration, and mood regulation [25]. Additionally, lisdexamfetamine has a similar action to methylphenidate, inhibiting the reuptake of these neurotransmitters by terminal neurons [24]. This increases the availability of catecholamines in the synapse, facilitating communication between neurons and aiding in cognitive control and behavior regulation [23]. Nonetheless, it commonly presents side effects such as appetite suppression, weight loss, insomnia, irritability, dizziness, possible growth reduction, exacerbation of tics, and increased blood pressure [26].

When it comes to non-stimulant medications, atomoxetine selectively inhibits the reuptake of norepinephrine in neurons, increasing its availability in the synapse [27]. Consequently, this leads to greater activation of noradrenergic receptors in the prefrontal cortex, which is involved in the regulation of attention, impulsivity, and cognitive control [28]. Synergistically, the inhibition of norepinephrine transport in the synaptic cleft increases both dopamine and norepinephrine levels in the prefrontal cortex, thereby addressing the two main neurotransmission differences observed in ADHD [23]. However, this medication also has potential side effects, such as headache, nausea, abdominal pain, decreased appetite, irritability, and drowsiness [29].

Guanfacine and clonidine, on the other hand, act on the α2 adrenergic receptors present in neuronal cells [30]. However, guanfacine is more specific for α2A receptors compared to clonidine, resulting in higher therapeutic efficacy with a reduced side effect profile [30]. The therapeutic benefits of guanfacine are related to increased functioning of the prefrontal cortex, which, through increased noradrenergic signaling, leads to improvements in behavior and impulsivity [31]. The main side effects of guanfacine include drowsiness, sedation, headache, abdominal pain, and fatigue [32]. Clonidine, in addition to drowsiness, can cause dry mouth and constipation as side effects, as well as hypotension and changes in heart rate [32].

Furthermore, modafinil is a stimulant that inhibits dopamine reuptake by binding to the dopamine transporter and, in the cortex, also inhibits norepinephrine reuptake [33]. Although less commonly used, this drug has evidence of improvement in ADHD symptoms [34]. However, it can also have side effects such as anxiety, headaches, nausea, decreased contentment, insomnia, stomach pain, or dry mouth [32].

In addition, antidepressants such as bupropion and certain tricyclics can also be used as off-label treatments for ADHD when first- or second-line medications are ineffective or produce intolerable side effects [35]. Similar to stimulants and atomoxetine, bupropion inhibits the reuptake of dopamine and norepinephrine. However, it has more potential side effects such as dizziness, tachycardia, anorexia, nausea, vomiting, irritability, hyperhidrosis, headache, insomnia, tremor, agitation, and anxiety [36]. Tricyclic antidepressants, on the other hand, are effective because they act on the reuptake of serotonin, norepinephrine, and dopamine [37]. Nevertheless, these medications not only have lower efficacy compared to stimulants but also present various side effects. These include arrhythmias, tachycardia, numbness, paresthesias of the extremities, tremors, dry mouth, blurred vision, urinary retention, constipation, nausea, weight gain, and erectile/ejaculation dysfunctions [32].

In general, these important side effects may be intolerable for some patients [38], leading to the discontinuation of therapy, which is a frequently encountered phenomenon [39]. Therefore, it may be beneficial to consider alternative treatments to the current pharmacological options used in ADHD. In this context, caffeine appears to increase the release of dopamine and norepinephrine in the synaptic cleft, but through competitive blockade of the adenosine receptor [40]. Furthermore, studies with animal models have shown that caffeine increases dopamine levels in the nucleus accumbens [41], a circuit responsible for motivation [42]. In ADHD, there is a decrease in the function of this reward pathway [43], and this alteration is associated with the primary symptoms of ADHD [44]. Similarly, there is also evidence in the literature that some psychostimulants increase dopamine in the reward pathway of the nucleus accumbens [45].

These characteristics, similar to those of first-choice medication for managing ADHD, raise the hypothesis that caffeine also plays a role in reducing impulsivity, improving sustained attention, and reducing hyperactivity. Recent literature reviews [46,47,48] indicate that caffeine may serve as a therapeutic option in managing patients with ADHD, although this is controversial in the literature. This study aimed to perform a systematic review and meta-analysis comparing the effect of caffeine versus placebo on ADHD symptoms in children.

## 2. Materials and Methods

The systematic review and meta-analysis were conducted and reported in accordance with the guidelines outlined in the Cochrane Collaboration Handbook for Systematic Review of Interventions and the Preferred Reporting Items for Systematic Reviews and Meta-Analysis (PRISMA) Statement [49,50]. The prospective meta-analysis protocol was registered on 24 December 2022 in the Prospective Register of Systematic Reviews (PROSPERO 2022; CRD42022383848).

### 2.1. Eligibility Criteria

The inclusion criteria were: (I) randomized placebo-controlled clinical trials; (II) patients with an established diagnosis of ADHD; (III) intervention involving oral administration of caffeine; (IV) clinical assessment with a scale, checklist, or test measuring any ADHD symptom; (V) age group 5 to 17 years old; and (VI) published in English. Non-randomized comparisons and studies without a placebo control group were excluded. Studies that met inclusion criteria but did not report quantitative outcomes were included only in the systematic review and not in the meta-analysis.

### 2.2. Search Strategy

The search was conducted on 1 December 2022, in the following databases: PubMed, Embase, and Cochrane Library, without any restrictions on publication date, using the following search strategy: [ADHD OR “Attention Deficit Hyperactivity Disorder” OR hyperactivity] AND [Caffeine OR “energy drink” OR coffee] AND [child OR children OR pediatric]. After removing duplicates, two independent reviewers (G.P. and M.M.G.M.) examined the title and abstract of each study identified in our search strategy. They applied the pre-established inclusion and exclusion criteria. Subsequently, the same procedure was employed in the evaluation of the full text of eligible studies. Discrepancies between reviewers were resolved by discussion until a consensus was reached. The literature search strategy is summarized in Figure 1.

### 2.3. Data Extraction

Two independent reviewers (G.P., M.M.G.M.) extracted the following data: (I) author and year; (II) study design; (III) number of patients; (IV) mean age of study participants; (V) caffeine dose; and (VI) time duration of caffeine administration; (VII) objective tests and checklists applied to measure the outcome. These include the ADHD symptom scales “Conners Rating Scale” (C.R.S.) and “Davids’ Hyperkinetic Rating Scale” (D.H.R.S.); and (VIII) the endpoints for each study.

The “Davids’ Hyperkinetic Rating Scale” (D.H.R.S.) is a tool that was developed to evaluate and aid in the identification and measurement of the symptoms related to attention deficit hyperactivity disorder (ADHD) [51]. This scale takes into consideration the scholar’s performance and six symptoms of ADHD: hyperactivity, short attention span, variability, impulsiveness, irritability, and explosiveness [52]. Each of the items ranges from one to six, with one meaning much less than most children and six meaning much more than most children [53]. The scale lacks a midpoint, “about same as most children”; this is one of the disadvantages of this scale [53]. All responses must classify the child as slightly less or more than most children. It has assigned points ranging from 1 to 6 for each of the six hyperkinesis items; all the points were summed to produce a score [53]. The score produced ranged from 6 to 36 [53]. It is used with a base score of 24 points or higher to identify hyperactive children [53].

The “Corners Ratting Scale” (C.R.S.) is also widely used to assess ADHD and other behavior issues in children and adolescents [54]. This measure has been factor analyzed into five variables: hyperactivity, conduct problem, inattentive-passive, tension-anxiety, and sociability [54]. Each factor’s score is the mean of its items (on a 4-point scale, 0–3) [53].

### 2.4. Quality Assessment

The risk of bias evaluation was conducted independently by two authors (G.P. and M.M.G.M.) using the revised Cochrane Risk of Bias 2 tool (RoB-2) [55] for the quality assessment of the randomized studies. Disagreements were resolved by consensus between the authors. The Cochrane RoB-2 (Risk of Bias 2) tool was designed to provide a standardized and transparent framework for assessing the quality and reliability of scientific studies [55]. RoB-2 evaluates various aspects of study design and conduct, including (1) adequate randomization, (2) blinding of participants and assessors, (3) handling of missing data due to loss to follow-up, (4) outcome selection, and (5) biases in outcome selection and detection [49].

In addition, for assessing the risk of bias in crossover studies, a set of criteria applied to randomized clinical trials was used in this analysis. This approach takes into consideration the potential biases related to treatment allocation, adequate blinding of participants and assessors, as well as appropriate analysis and reporting of results. Disagreements were resolved by consensus between the authors.

### 2.5. Statistical Analysis

We compared treatment effects on the continuous outcome of overall symptoms using Standardized Mean Difference (SMD) with a 95% confidence interval. If the same ADHD outcome was evaluated by more than one person (e.g., parent and teacher), we combined the mean and standard deviation following the formula described in the Cochrane Handbook of Systematic Reviews of Interventions version 6.3 [49]. Heterogeneity was assessed using the Cochran Q test and I^2^ statistics. Pooled analyses were performed with a random effects model, as per recommendations from the Cochrane Collaboration [49]. Statistical analysis was performed using Review Manager 5.3 software (Cochrane Collaboration in Denmark).

## 3. Results

### 3.1. Study Selection and Characteristics

A total of 344 articles were found using the proposed search strategy. After the removal of duplicate records and ineligible studies, 12 studies remained and were fully reviewed based on the inclusion and exclusion criteria. Of these, a total of seven studies met all the inclusion criteria and were selected for the systematic review. Of these, four studies (*n* = 76) presented quantitative comparisons between the caffeine x placebo groups and were thus eligible for the meta-analysis.

The seven included studies were randomized placebo-controlled crossover trials, totaling 104 patients from 5 to 15 years old of both sexes. The caffeine doses administered ranged from 75 mg to 308.6 mg. The studies included in our review did not use children’s age to personalize the caffeine dosage. Most studies [54,56,57,58,59] used standard caffeine doses for children, regardless of age. Arnold et al. [52] personalized the caffeine dosage for each child weeks before the beginning of the study. The study group of Kahathuduwa et al. [56] used the children’s body weight as a measure, administering 2 mg of caffeine per kilogram of body weight. The caffeine administration period varied from 1 day to 3 weeks. The majority of studies [52,54,57] used capsules for caffeine supplementation, with only one study [58] using coffee. Two other studies used hot beverages for dose administration [59,60], and one study [56] used 100 mL of a solution to dilute the supplements. Regarding caffeine’s side effects, in two of the studies included in our review [56,59], none of the participants reported side effects. In another study [54], several children mentioned experiencing slight discomfort. Arnold et al. [52] observed weight loss in some individuals, while the other three studies [57,58,60] did not mention any side effects. Study characteristics and the main results of the individual studies are reported in Table 1.

### 3.2. Overall Symptoms

Five of the analyzed studies [52,54,57,59,60] included a general symptom range. Three [54,57,59] of them did not identify any difference in general ADHD symptoms, regardless of the scale applied. Arnold et al. [52] applied five checklists to evaluate general symptoms after the period of caffeine administration in patients, with a significant improvement result only in the “Parents symptom checklist reported”. Garfinkel et al. [60] applied a “Conners’ Teacher Rating Scale” in their study and did not find a significant improvement in the general symptoms of patients with ADHD at a dose of 308.6 mg; however, there was a significant improvement in symptoms with a dose of 158.6 mg.

A meta-analysis was performed to compare general ADHD symptoms (four studies; *n* = 76) [52,54,57,59]. There was no significant difference between caffeine and placebo in the incidence of overall ADHD symptoms (SMD: −0.12; 95% CI: −0.44 to 0.20; *p* = 0.45; I^2^ = 0%) (Figure 2).

### 3.3. Inattention

Five studies [52,56,58,59,60] reported inattention as an outcome. Three of these studies [52,58,59] did not show any significant change in inattention with caffeine compared with placebo. Garfinkel et al. [60], on the other hand, reported a significant improvement when assessing inattention using the Conners Rating Scale (Inattentiveness-Factor II) in patients who received 158.6 mg of caffeine. However, there was no significant improvement when the same patients received 308.6 mg of caffeine. Kahathuduwa et al. [56] reported better accuracy in the “Go/NoGo” task of patients under the effects of caffeine, indicating an improvement in sustained attention.

### 3.4. Hyperactivity

Only three studies [52,59,60] presented the outcome of hyperactivity. Two of them [52,59] did not show significant improvement in hyperactivity when comparing caffeine with placebo. However, Garfinkel et al. [60] also described a significant improvement when assessing inattention with the “Conners’ Rating Scale (Hyperactivity-Factor IV)” in patients who received 158.6 mg of caffeine, but no significant difference between placebo and caffeine when these same patients received 308.6 mg of caffeine.

### 3.5. Impulsivity

Three studies [52,54,56] presented impulsivity as an outcome. Two of them [52,54] did not show significant changes in impulsivity when comparing caffeine with placebo. Kahathuduwa et al. [56], however, applied the Stop signal task and described a significant decrease in inhibitory control in patients using caffeine compared with patients using placebo, indicating an increase in impulsivity in this sample.

### 3.6. Risk of Bias

We performed a risk of bias assessment with the RoB-2 tool [55]. The assessment of each randomized controlled trial is reported in Table 2. From the analyzed studies, one study presented a low risk of bias [56]. The bias analysis algorithm categorized the remaining six articles [52,54,57,58,59,60] as “some concerns.” The study protocols for these six articles were not found by the authors, leading to a determination of “some concerns” regarding the selection of the reported outcomes.

## 4. Discussion

Caffeine is categorized as a central nervous system stimulant and is composed of methylxanthine [61]. Its main mechanism of action is the antagonism of adenosine receptors, which, consequently, increases dopaminergic actions in the cortex [62]. In addition to these dopaminergic effects, caffeine produces secondary effects on acetylcholine and noradrenaline [63].

In adults without any diagnosed mental disorders, caffeine at doses up to approximately 300 mg improves cognitive processes such as reaction time and attention with minimal side effects [64]. Some less consistent effects have also been noted for executive function, judgment, and decision-making, as well as memory [64]. Another study also showed improved sustained attention in adults without psychiatric diagnoses after caffeine use [65]. Specifically in the sports context, a literature review suggested that the ingestion of low to moderate doses of caffeine can boost energy before or during exercise, attention, mood, and self-reported cognitive functions [66]. Depending on the research method, it might also enhance memory, weariness, choice reaction time, and simple reaction time [66].

Furthermore, caffeine and caffeine-containing beverages, such as coffee and tea, appear to act as protective factors when consumed regularly [67,68]. Coffee consumption appears to reduce the risk of depression, with the most significant protective impact observed at 400 mL/day [67]. Additionally, cross-sectional and longitudinal population-based investigations also imply that coffee, tea, and caffeine alone may protect against cognitive impairment/decline in later life. However, this link was not consistently observed across all cognitive domains that were tested [68].

Several recent studies in animal models [69,70,71] have shown that caffeine can be an adjuvant in the treatment of ADHD. Vázquez et al. [46] conducted a systematic review on the effects of caffeine on symptoms similar to ADHD in animal models. This suggests that caffeine is a possible pharmacological adjuvant for treating ADHD and raises the hypothesis caffeine’s cognitive effects in animal models could be applied to ADHD patients.

However, our review showed that, at least in the current literature, there are few signs that caffeine improves the general symptoms of ADHD. Three of the five studies [54,57,59] reporting this outcome did not show a significant difference between children using caffeine relative to placebo. Nonetheless, one of the trials [52] found equivocal results, with improvement in only one of five scales, and the other showed a benefit only with a reduced dose of caffeine [60]. Moreover, our meta-analysis with four studies did not find any evidence of improvement in general ADHD symptoms with caffeine relative to placebo. Importantly, the results of these four studies were also consistent with each other, with low heterogeneity between studies (I^2^ = 0%).

The systematic review also showed no benefit in the outcome of impulsivity in two studies, with evidence of harm (increased impulsivity) by caffeine in one study [56]. Similarly, inattention and hyperactivity were also not discernible differences between groups in the majority of studies. In contrast, Garfinkel et al. [60] showed that lower doses of caffeine (158.6 mg/day) were associated with improvement in inattention and hyperactivity in patients with ADHD, an effect that was not seen with a higher dose. Notably, two other studies [58,59] that also used lower doses (75–180 mg) did not find any evidence in favor of caffeine. Significantly, differences in the literature may also be related to the small number of participants in some studies, as low as six and eight children [59,60].

Some of the studies included in our review [52,54,57,59,60] also compared caffeine with methylphenidate, one of the first-line medications currently used in the treatment of ADHD. In the study by Garfinkel et al. [60], where caffeine at a dose of 158.6 mg appeared to improve symptoms of inattention and hyperactivity, this substance had a much more subtle effect compared to methylphenidate. Furthermore, it is noticeable that caffeine has a half-life of 3 to 5 h [64], providing cognitive effects for up to 3 h [72], similar to methylphenidate, which has a duration of action of 1 to 4 h and a half-life of 2 to 3 h [73].

Non-randomized studies on the effects of caffeine use in patients with ADHD have similarly shown conflicting results. A study involving 25 children reported that caffeine did not help any patient and in fact worsened symptoms in some of them [74]. In contrast, a comparison of 12 children treated with caffeinated vs. decaffeinated coffee significantly improved general ADHD symptoms [75]. The controversial effect of caffeine on patients with ADHD is also seen in the literature for the adult population. A cross-sectional study involving 2259 adult patients with ADHD concluded that caffeine does not seem to be successful for self-medication of ADHD symptoms [76]. In contrast, a study analyzed data from 1239 soldiers with ADHD and showed that those who consumed caffeine had less impulsive behavior [77].

Comparatively analyzing the results of caffeine with off-label medications for ADHD, it is found that antidepressants appear to be more promising than caffeine. A meta-analysis with a similar methodology compared placebo with tricyclic antidepressants in children with ADHD [78]. The results showed that tricyclic antidepressants were more effective than placebo in the three included randomized clinical trials (OR 18.50; 95% CI 6.29 to 54.39; I^2^ = 1%) [78]. This finding was consistent for both tricyclic antidepressants (desipramine and nortriptyline) [78].

A network meta-analysis compared amphetamines, atomoxetine, bupropion, clonidine, guanfacine, methylphenidate, and modafinil in the treatment of core symptoms of ADHD in children and adolescents [79]. In this study, all included medications were superior to placebo when rated by physicians. However, when rated only by teachers, only methylphenidate and modafinil outperformed placebo [79]. Bupropion had significant efficacy compared to placebo when evaluated by physicians (OR −0.96; 95% CI −1.69 to −0.22), although according to this trial, it did not show a significant difference from placebo when evaluated by teachers (OR −0.32; 95% CI −1.07 to 0.43) [79].

Some recent studies [80,81,82,83] show that there are more promising options for daily substances found in food that can assist in managing ADHD. Some dietary interventions may have a small, but significant, effect on ADHD symptoms. Emerging evidence highlights the therapeutic potential of prebiotics and probiotics in addressing ADHD symptoms [80,81]. Additionally, a noteworthy observation is that dietary supplementation with zinc and iron corresponds with improvements in ADHD severity [82]. Furthermore, exclusion of artificial food colors and supplementation of free fatty acids, particularly omega-3 and omega-6, showed promise in reducing ADHD symptoms [83].

The limitations of this systematic review and meta-analysis should be acknowledged. First, it should be noted that there are few randomized controlled trials in the literature on this topic and that each study involves a small number of patients. Consequently, there is a vital need for more trials to enhance the evidential basis. Nevertheless, our meta-analysis represents the largest aggregate population comparing caffeine vs. placebo in a randomized pediatric population with ADHD. Second, the studies used different tools for outcome assessment. Therefore, we used a standardized mean difference for comparison of the continuous outcome between groups. Third, there were different doses of caffeine between studies.

## 5. Conclusions

The totality of evidence in randomized controlled trials comparing caffeine with placebo for treatment of ADHD symptoms in children suggests that there is likely no benefit of caffeine for management of general symptoms such as hyperactivity, inattention, and impulsivity in this patient population. For the advancement of ADHD therapy, it will be of greater importance to seek new molecules for the treatment of this disorder.

## Figures and Tables

**Figure 1 brainsci-13-01304-f001:**
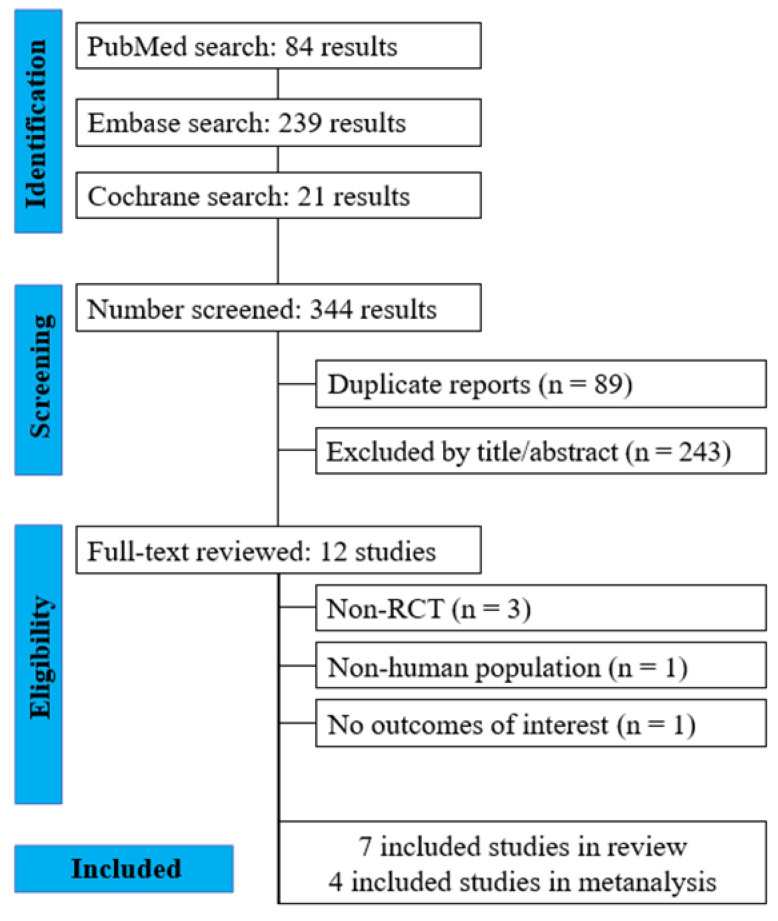
PRISMA flow diagram of study screening and selection.

**Figure 2 brainsci-13-01304-f002:**
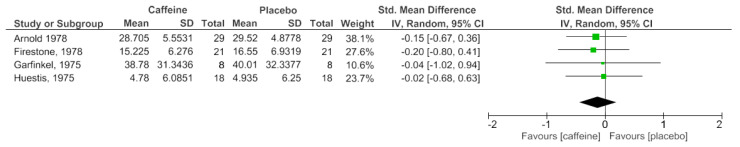
Forest plot comparing caffeine and placebo in ADHD overall symptoms [52,54,57,59].

**Table 1 brainsci-13-01304-t001:** Summary of included studies.

Author and Year	Patients (n)/Sex (Male/Female)	Range Age (Years)	Caffeine Dose (mg)	Time of Use (Days)	Behavior Tests and Checklists	Main Results
Arnold, 1978 [52]	29 (22/7)	5–12	160–300	21	– Conners Teachers Behavior Problem Checklist. – Davids Hyperkinetic Rating Scale was completed by the parents and by the teacher. – Parents symptom checklist reported. – Target symptom assessment by parents with the aid of the psychiatrist.	– They all show essentially the same pattern: caffeine was not significantly better than placebo. – Caffeine outperformed placebo solely on the problem checklist provided by parents (*p* < 0.05).
Conners, 1979 [58]	17 (17/0)	8–11	75–180	1	– Continuous Performance Test (CPT).	– There was no significant difference in commission errors or errors of omission when comparing caffeine and placebo.
Firestone, 1978 [54]	21 (NA)	6–12	300–500	21	– Matching Familiar Figures Test (MFF). – The Maze Test. – Conners Rating Scale (CRS) by mothers and by teachers. – Conners Short Form Rating Scale (CSS). – Reaction-Time Apparatus.	– In the behavioral scales answered by mothers and teachers, placebo and caffeine were not different. – The frequency of impulsive responses also did not differ between placebo and caffeine during the tests.
Garfinkel, 1975 [59]	8 (NA)	6–10	150	10	– Conners Teacher Rating Scale. – Bender Visual Motor Gestalt Test. – Frostig Developmental Test of Visual Perception, parts II and IV. – Reitan Neuropsychological Battery Test for Motor Coordination and Steadiness. – Kagan Matching Familiar Figures Test.	– Total scores of general symptoms during the caffeine period were not statistically different from scores during the baseline and placebo periods. – The caffeine period aggressiveness, hyperactivity, inattention, and sociability scores were, again, not significantly different from those obtained during the baseline and placebo periods. – Caffeine did not significantly improve scores on any of the factors examined by the complete set of psychological tests.
Garfinkel, 1981 [60]	6 (6/0)	6–10	158.6–308.6	21	– Conners Teacher Rating Scale.	– High-dose caffeine (308.6 mg) did not differ from placebo. – Caffeine at the low dose (158.6 mg) outperformed placebo (*p* < 0.01) and caffeine at the high dose (*p* < 0.05). – The observed effect of caffeine to enhance behavior at 158.6 mg was significant for aggressiveness, inattentiveness, and hyperactivity.
Huestis, 1975 [57]	18 (12/6)	5–12	80–300	21	– Davids Hyperkinetic Rating Scale, completed by the parents and by the teacher. – Parents symptom checklist reported. – Conners Teachers Behavior Checklist. – Target symptom assessment by parents with the aid of the psychiatrist.	– Data from that study show that caffeine was not significantly more effective than placebo in any of the tests.
Kahathuduwa, 2020 [56]	5 (5/0)	8–15	56–210	1	– NIH Cognition Toolbox. – Go/NoGo task. – Stop signal task.	– In the NIH Cognition Toolbox Test Battery, caffeine improved the total cognition composite compared to placebo; however, this difference was not statistically significant. (*p* = 0.148). – In the Go/NoGo task, compared to the placebo, caffeine was connected to an improvement trend in the sensitivity to the Go signal (*p* = 0.057). Caffeine significantly improved the Go/NoGo hit rate but did not have a significant effect on the reaction times or the false alarm rates in the Go/NoGo task. – In the Stop signal task, caffeine did not significantly change the reaction times to the Go stimuli compared to placebo, but administration of caffeine was associated with decreased inhibitory control (*p* = 0.031) compared to placebo.

**Table 2 brainsci-13-01304-t002:** Risk of bias summary for randomized studies (RoB-2) for crossover trials.

Study	Bias from Randomization Process	Bias Arising from Period and Carryover Effects	Bias due to Deviations from Intended Interventions	Bias due to Missing Outcome Data	Bias in Measurement of the Outcomes	Bias in Selection of the Reported Result	Overall Risk of Bias
Arnold, 1978 [52]	Low	Low	Low	Low	Low	Some concerns	Some concerns
Conners, 1979 [58]	Low	Low	Low	Low	Low	Some concerns	Some concerns
Firestone, 1978 [54]	Low	Low	Low	Low	Low	Some concerns	Some concerns
Garfinkel, 1975 [59]	Low	Low	Low	Low	Low	Some concerns	Some concerns
Garfinkel, 1981 [60]	Low	Low	Low	Low	Low	Some concerns	Some concerns
Huestis, 1975 [57]	Low	Low	Low	Low	Low	Some concerns	Some concerns
Kahathuduwa, 2020 [56]	Low	Low	Low	Low	Low	Low	Low

## Data Availability

Not applicable.

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
