# Peer review of "Effects of Caffeine on Main Symptoms in Children with ADHD: A Systematic Review and Meta-Analysis of Randomized Trials"

_brainsci, 2023, doi:10.3390/brainsci13091304_

Round 1

Reviewer 1 Report

This is a very interesting study, pointing to a novel and critical topic. However, in my opinion, the paper has shortcomings regarding methodology and rationale that need to be considered:

1.     The type of symptoms need to be clarified in the title. (Is it all the symptoms? General symptoms? Specific groups of symptoms?) it is a very broad range. 

2.     Could you please provide clarification on the meaning of "4In adults" in Line 50?

3.     The rationale behind considering caffeine as an alternative option for ADHD medication lacks clarity. 

4.     What is the magnitude of caffeine's impact on dopamine and norepinephrine release?

5.     How does this compare to the effects of standard ADHD medications? 

6.     Is there a specific dosage of caffeine that can mimic the effects of ADHD medication? 

7.     Is this dosage different for different age groups among pediatrics?

8.     The pediatric age group and target population need to be defined. 

9.     Why was the age group not included in the eligibility criteria?

10.  Are there any gender differences that need to be considered? It's essential for the authors to clarify whether gender plays a significant role in the context of this study.

11.  What is the duration of caffeine's effects per serving size, especially when compared to ADHD medication?

12.  What are the side effects if of caffeine consumption for the target age group? 

13.  What was the form of caffeine supplementation in the included studies? (Drinks, capsules, gel, sachet?). 

14.  A similarity in the text was found. Please rewrite and paraphrase accordingly (mainly in section 3.6 and discussion) 

15.  The pool of available studies is limited and more studies need to be included for reliability of results. 

16.  It is recommended to mention the general effectiveness of dietary compounds on ADHD disorder (sugar, probiotics, dietary fiber, etc..) on example could be: https://www.emerald.com/insight/content/doi/10.1108/NFS-12-2021-0388/full/html?skipTracking=true&utm_campaign=Emerald_Health_PPV_Dec22_RoN Probiotic supplement as adjunctive therapy with Ritalin for treatment of attention-deficit hyperactivity disorder symptoms in children: a double-blind placebo-controlled randomized clinical trial

The quality of the English language is acceptable except for some minor edits in the test. 

Author Response

August 18, 2023

Re: Effects of Caffeine on Symptoms in Children with ADHD: A Systematic Review and Meta-Analysis of Randomized Trials

Giuliana Perrotte, Marina Gonzaga Moreira, Amauri de Vargas Junior, Alvaro Teixeira Filho and João Mauricio Castaldelli-Maia

Dear Editor,

We are very appreciative of the careful reading which you gave our manuscript. We meticulously addressed each of the points and questions raised and performed the corrections or justifications in accordance. In the discussion that follows we have indicated our direct responses to the questions raised by your thorough review. We believe the manuscript significantly improved because of your constructive criticisms and hope that you will find the revised version of this manuscript acceptable.

Point 1.     The type of symptoms need to be clarified in the title. (Is it all the symptoms? General symptoms? Specific groups of symptoms?) it is a very broad range.

Response 1: Thank you for this comment. Even though our article specifies in the abstract and in the results section that the investigated symptoms were overall symptoms, inattention, hyperactivity, and impulsivity, now we have added "main" to the title, and avoided excessive length.

Point 2.     Could you please provide clarification on the meaning of "4In adults" in Line 50?

Response 2: Thank you for bringing this to our attention. The number 4 appears that there was an error in line 50. The correct sentence is “In adults, a minimum of five symptoms in at least one of the three domains is required for diagnosis.”

Point 3.     The rationale behind considering caffeine as an alternative option for ADHD medication lacks clarity.

Response 3: Dear reviewer, we sincerely appreciate your constructive criticism, which is instrumental in striving for excellence in our work. We describe further in this matter in manuscript on lines 131 to 136.

Point 4.     What is the magnitude of caffeine's impact on dopamine and norepinephrine release?

Response 4: We didn’t find this information specifically in the literature. However, in the Vazquez review [1], who compiled studies involving animals, an increase in dopamine was observed in the striata region of the brain in animals that had consumed caffeine.

[1] Vázquez, Javier C et al. “Effects of Caffeine Consumption on Attention Deficit Hyperactivity Disorder (ADHD) Treatment: A Systematic Review of Animal Studies.” Nutrients vol. 14,4 739. 10 Feb. 2022, doi:10.3390/nu14040739

Point 5.     How does this compare to the effects of standard ADHD medications?

Response 5: Thank you for your inquiries; they prompted us to reflect and incorporate additional points into our review. Some of the studies included in our research [1-5] also compare caffeine with methylphenidate, one of the first-line medications in the current treatment of ADHD. Across all the studies [1-5], caffeine was less effective in reducing ADHD symptoms as measured by the applied scales. In the study by Garfinkel et al. [4], caffeine at a dose of 158.6mg appeared to improve overall symptoms, inattention, and hyperactivity in the 6 children included in the study, but still to a lesser extent than methylphenidate. We have included this information in the discussion (lines 334 to 338).

[1] Arnold, L E et al. “Methylphenidate vs dextroamphetamine vs caffeine in minimal brain dysfunction: controlled comparison by placebo washout design with Bayes' analysis.” Archives of general psychiatry vol. 35,4 (1978): 463-73. doi:10.1001/archpsyc.1978.01770280073008

[2] Firestone, P et al. “The effects of caffeine and methylphenidate on hyperactive children.” Journal of the American Academy of Child Psychiatry vol. 17,3 (1978): 445-56. doi:10.1016/s0002-7138(09)62300-1

[3] Garfinkel, B D et al. “Methylphenidate and caffeine in the treatment of children with minimal brain dysfunction.” The American journal of psychiatry vol. 132,7 (1975): 723-8. doi:10.1176/ajp.132.7.723

[4] Garfinkel, B D et al. “Responses to methylphenidate and varied doses of caffeine in children with attention deficit disorder.” Canadian journal of psychiatry. Revue canadienne de psychiatrie vol. 26,6 (1981): 395-401. doi:10.1177/070674378102600602

[5] Huestis, R D et al. “Caffeine versus methylphenidate and d-amphetamine in minimal brain dysfunction: a double-blind comparison.” The American journal of psychiatry vol. 132,8 (1975): 868-70. doi:10.1176/ajp.132.8.868

Point 6.     Is there a specific dosage of caffeine that can mimic the effects of ADHD medication?

Response 6: Thank you for this question. Only the study by Garfinkel et al. proposes that a specific dosage of caffeine can mimic the effects of medication for add. In this study comparing methylphenidate with caffeine and placebo, the author suggests that caffeine at low doses (158.6mg) may improve ADHD symptoms, while higher dosages (308.6mg) would not be helpful. However, other groups of authors [2-3] also used low doses of caffeine and did not demonstrate an improvement in symptoms when compared to placebo. It is important to highlight that these two studies [2-3] also have larger sample sizes than those of Garfinkel et al.

[1] Garfinkel, B D et al. “Responses to methylphenidate and varied doses of caffeine in children with attention deficit disorder.” Canadian journal of psychiatry. Revue canadienne de psychiatrie vol. 26,6 (1981): 395-401. doi:10.1177/070674378102600602

[2] Garfinkel, B D et al. “Methylphenidate and caffeine in the treatment of children with minimal brain dysfunction.” The American journal of psychiatry vol. 132,7 (1975): 723-8. doi:10.1176/ajp.132.7.723

[3] Conners, C K. “The acute effects of caffeine on evoked response, vigilance, and activity level in hyperkinetic children.” Journal of abnormal child psychology vol. 7,2 (1979): 145-51. doi:10.1007/BF00918895

Point 7.     Is this dosage different for different age groups among pediatrics?

Response 7: Thank you for your valuable input. The studies included in our review did not use the age of the children to personalize the caffeine dosage. Kahathuduwa et al. [1] used the children's body weight as a measure, administering 2.0 mg of caffeine per kilogram of body weight. The study group of Arnold et al. [2] personalized the caffeine dosage for each child weeks before the study began. The other 5 studies [3-7] included utilized standard doses for the children, regardless of their age. We have included this passage in the results section of the manuscript (lines 224 to 229).

[1] Kahathuduwa, Chanaka N et al. “Effects of L-theanine-caffeine combination on sustained attention and inhibitory control among children with ADHD: a proof-of-concept neuroimaging RCT.” Scientific reports vol. 10,1 13072. 4 Aug. 2020, doi:10.1038/s41598-020-70037-7

[2] Arnold, L E et al. “Methylphenidate vs dextroamphetamine vs caffeine in minimal brain dysfunction: controlled comparison by placebo washout design with Bayes' analysis.” Archives of general psychiatry vol. 35,4 (1978): 463-73. doi:10.1001/archpsyc.1978.01770280073008

[3] Conners, C K. “The acute effects of caffeine on evoked response, vigilance, and activity level in hyperkinetic children.” Journal of abnormal child psychology vol. 7,2 (1979): 145-51. doi:10.1007/BF00918895

[4] Firestone, P et al. “The effects of caffeine and methylphenidate on hyperactive children.” Journal of the American Academy of Child Psychiatry vol. 17,3 (1978): 445-56. doi:10.1016/s0002-7138(09)62300-1

[5] Garfinkel, B D et al. “Methylphenidate and caffeine in the treatment of children with minimal brain dysfunction.” The American journal of psychiatry vol. 132,7 (1975): 723-8. doi:10.1176/ajp.132.7.723

[6] Garfinkel, B D et al. “Responses to methylphenidate and varied doses of caffeine in children with attention deficit disorder.” Canadian journal of psychiatry. Revue canadienne de psychiatrie vol. 26,6 (1981): 395-401. doi:10.1177/070674378102600602

[7] Huestis, R D et al. “Caffeine versus methylphenidate and d-amphetamine in minimal brain dysfunction: a double-blind comparison.” The American journal of psychiatry vol. 132,8 (1975): 868-70. doi:10.1176/ajp.132.8.868

Point 8.     The pediatric age group and target population need to be defined.

Response 8: Thank you for bringing this to our attention. We included studies in our review that targeted children aged 5 to 17 years with a diagnosis of ADHD.

Point 9.     Why was the age group not included in the eligibility criteria?

Response 9: We apologize for the oversight of not describing age as an eligibility criterion in the methods section. This criterion was used for our analysis; however, it was inadvertently omitted when describing the methods. Our inclusion criteria were children aged 5 to 17 years, and this information has been added to the manuscript on line 152.

Point 10.  Are there any gender differences that need to be considered? It's essential for the authors to clarify whether gender plays a significant role in the context of this study.

Response 10: Thank you very much for your input. We mistakenly omitted the patients' gender because this information was not available in all the articles included in the review. Following your insightful suggestion, we have added this information to Table 1.

Point 11.  What is the duration of caffeine's effects per serving size, especially when compared to ADHD medication?

Response 11: Thank you for your question. The duration of caffeine's effect is not addressed in any of the studies included in our review. However, literature data [1,2] show that the cognitive effects of caffeine last from 3 to 5 hours, which is a duration of effect very close to that of methylphenidate (around 4 hours).

[1] McLellan, Tom M et al. “A review of caffeine's effects on cognitive, physical and occupational performance.” Neuroscience and biobehavioral reviews vol. 71 (2016): 294-312. doi:10.1016/j.neubiorev.2016.09.001

[2] Institute of Medicine (US) Committee on Military Nutrition Research; Marriott BM, editor. Food Components to Enhance Performance: An Evaluation of Potential Performance-Enhancing Food Components for Operational Rations. Washington (DC): National Academies Press (US); 1994. 20, Effects of Caffeine on Cognitive Performance, Mood, and Alertness in Sleep-Deprived Humans. Available from: https://www.ncbi.nlm.nih.gov/books/NBK209050/

Point 12.  What are the side effects if of caffeine consumption for the target age group?

Response 12: Thank you for bringing this to our attention. In two of the included studies [1,2], none of the participants reported side effects. In one study [3], several children mentioned experiencing slight discomfort. In another study [4], weight loss was observed in some individuals. The other three studies [5-7] did not mention any side effects. Literature reports indicate common side effects of caffeine use in children, including headaches, sleep disturbances, and fatigue [8]. We have included the results regarding side effects in the manuscript (lines 232 to 236).

[1] Garfinkel, B D et al. “Methylphenidate and caffeine in the treatment of children with minimal brain dysfunction.” The American journal of psychiatry vol. 132,7 (1975): 723-8. doi:10.1176/ajp.132.7.723

[2] Kahathuduwa, Chanaka N et al. “Effects of L-theanine-caffeine combination on sustained attention and inhibitory control among children with ADHD: a proof-of-concept neuroimaging RCT.” Scientific reports vol. 10,1 13072. 4 Aug. 2020, doi:10.1038/s41598-020-70037-7

[3] Firestone, P et al. “The effects of caffeine and methylphenidate on hyperactive children.” Journal of the American Academy of Child Psychiatry vol. 17,3 (1978): 445-56. doi:10.1016/s0002-7138(09)62300-1

[4] Arnold, L E et al. “Methylphenidate vs dextroamphetamine vs caffeine in minimal brain dysfunction: controlled comparison by placebo washout design with Bayes' analysis.” Archives of general psychiatry vol. 35,4 (1978): 463-73. doi:10.1001/archpsyc.1978.01770280073008

[5] Huestis, R D et al. “Caffeine versus methylphenidate and d-amphetamine in minimal brain dysfunction: a double-blind comparison.” The American journal of psychiatry vol. 132,8 (1975): 868-70. doi:10.1176/ajp.132.8.868

[6] Conners, C K. “The acute effects of caffeine on evoked response, vigilance, and activity level in hyperkinetic children.” Journal of abnormal child psychology vol. 7,2 (1979): 145-51. doi:10.1007/BF00918895

[7] Garfinkel, B D et al. “Responses to methylphenidate and varied doses of caffeine in children with attention deficit disorder.” Canadian journal of psychiatry. Revue canadienne de psychiatrie vol. 26,6 (1981): 395-401. doi:10.1177/070674378102600602

[8] Soós R, Gyebrovszki Á, Tóth Á, Jeges S, Wilhelm M. Effects of Caffeine and Caffeinated Beverages in Children, Adolescents and Young Adults: Short Review. Int J Environ Res Public Health. 2021 Nov 25;18(23):12389. doi: 10.3390/ijerph182312389. PMID: 34886115; PMCID: PMC8656548.

Point 13.  What was the form of caffeine supplementation in the included studies? (Drinks, capsules, gel, sachet?).

Response 13: Thank you for bringing this to our attention. Most of the studies [1-3] used capsules for caffeine supplementation, with only one study [4] using coffee. Two other studies used hot beverages for dose administration [5-6], and one study [7] used a 100ml solution to dilute the supplements. We have added this important information to the results in the manuscript (lines 230 to 232).

[1] Arnold, L E et al. “Methylphenidate vs dextroamphetamine vs caffeine in minimal brain dysfunction: controlled comparison by placebo washout design with Bayes' analysis.” Archives of general psychiatry vol. 35,4 (1978): 463-73. doi:10.1001/archpsyc.1978.01770280073008

[2] Firestone, P et al. “The effects of caffeine and methylphenidate on hyperactive children.” Journal of the American Academy of Child Psychiatry vol. 17,3 (1978): 445-56. doi:10.1016/s0002-7138(09)62300-1

[3] Huestis, R D et al. “Caffeine versus methylphenidate and d-amphetamine in minimal brain dysfunction: a double-blind comparison.” The American journal of psychiatry vol. 132,8 (1975): 868-70. doi:10.1176/ajp.132.8.868

[4] Conners, C K. “The acute effects of caffeine on evoked response, vigilance, and activity level in hyperkinetic children.” Journal of abnormal child psychology vol. 7,2 (1979): 145-51. doi:10.1007/BF00918895

[5] Garfinkel, B D et al. “Methylphenidate and caffeine in the treatment of children with minimal brain dysfunction.” The American journal of psychiatry vol. 132,7 (1975): 723-8. doi:10.1176/ajp.132.7.723

[6] Garfinkel, B D et al. “Responses to methylphenidate and varied doses of caffeine in children with attention deficit disorder.” Canadian journal of psychiatry. Revue canadienne de psychiatrie vol. 26,6 (1981): 395-401. doi:10.1177/070674378102600602

[7] Kahathuduwa, Chanaka N et al. “Effects of L-theanine-caffeine combination on sustained attention and inhibitory control among children with ADHD: a proof-of-concept neuroimaging RCT.” Scientific reports vol. 10,1 13072. 4 Aug. 2020, doi:10.1038/s41598-020-70037-7

Point 14.  A similarity in the text was found. Please rewrite and paraphrase accordingly (mainly in section 3.6 and discussion)

Response 14:  Thank you for bringing this to our attention. We have rephrased the passages similar to those of other authors.

Point 15.  The pool of available studies is limited and more studies need to be included for reliability of results. 

Response 15: We appreciate this insightful observation. Despite conducting a comprehensive search across databases, the current literature presents a significant gap in the number of available studies on the use of caffeine in children with ADHD. Unfortunately, this has resulted in a limited amount of data for the review when considering only randomized clinical trials. We believe it's important to include only this type of study due to the higher level of evidence provided by this design.

Point 16.  It is recommended to mention the general effectiveness of dietary compounds on ADHD disorder (sugar, probiotics, dietary fiber, etc..) on example could be: https://www.emerald.com/insight/content/doi/10.1108/NFS-12-2021-0388/full/html?skipTracking=true&utm_campaign=Emerald_Health_PPV_Dec22_RoN Probiotic supplement as adjunctive therapy with Ritalin for treatment of attention-deficit hyperactivity disorder symptoms in children: a double-blind placebo-controlled randomized clinical trial      

Response 16: Thank you very much for your brilliant observation. We have added a paragraph on this topic in the discussion (lines 364 to 371).

Reviewer 2 Report

This manuscript entitled “Effects of Caffeine on Symptoms in Children with ADHD: A Systematic Review and Meta-Analysis of Randomized Trials” investigated the effects of caffeine consumption on attention-deficit/hyperactivity disorder’s symptoms in compare with placebo. The topic is interesting; however, the manuscript needs some editions. 

1.      Punctuation needs revision, the sentences must be finished like “being 4 times more affected than girls [2]. not being 4 times more affected than girls. [2]

2.      Line 59, it seems number 4 in beginning of the sentence is a mistake, isn’t it?

3.      The introduction is written in a lot of detail and is better to summarize.

4.      On page 6 (table 1), it is better to repeat the table headers for the convenience of the readers.

Punctuation needs revision, the sentences must be finished like “being 4 times more affected than girls [2]. not being 4 times more affected than girls. [2]

Author Response

August 18, 2023

Re: Effects of Caffeine on Symptoms in Children with ADHD: A Systematic Review and Meta-Analysis of Randomized Trials

Giuliana Perrotte1, Marina Gonzaga Moreira, Amauri de Vargas Junior, Alvaro Teixeira Filho and João Mauricio Castaldelli-Maia

Dear Editor,

We are very appreciative of the careful reading which you gave our manuscript. We meticulously addressed each of the points and questions raised and performed the corrections or justifications in accordance. In the discussion that follows we have indicated our direct responses to the questions raised by your thorough review. We believe the manuscript significantly improved because of your constructive criticisms and hope that you will find the revised version of this manuscript acceptable.

Point 1. Punctuation needs revision, the sentences must be finished like “being 4 times more affected than girls [2]. not being 4 times more affected than girls. [2]

Response 1: Thank you very much for your correction. All punctuation has been reviewed and corrected in the manuscript.

Point 2.  Line 59, it seems number 4 in beginning of the sentence is a mistake, isn’t it?

Response 2: Thank you for bringing this to our attention. You are correct, it appears that there was an error in the third paragraph of the Introduction. The correct sentence is “In adults, a minimum of five symptoms in at least one of the three domains is required for diagnosis.”

Point 3.  The introduction is written in a lot of detail and is better to summarize.

Response 3: Thank you for this comment. We have condensed some repetitive sections in the introduction.

Point4. On page 6 (table 1), it is better to repeat the table headers for the convenience of the readers.

Response 4: Thank you very much for your brilliant suggestion. We have made changes to the manuscript and added table headers to Table 1 whenever there is a page change.

Reviewer 3 Report

This is a well-structured systematic review. The main question addressed by this paper is the effects of caffeine on symptoms in children with attention-deficit/hyperactivity disorder (ADHD).

The introduction gives the background of this study as it briefly describes ADHD, its definition, clinical presentations, diagnosis and treatment.

“Materials and Methods” section is descriptive enough. It refers to the eligibility criteria, search strategy, data extraction, quality assessment and statistical analyses implemented during the meta-analysis.

The results are quite interesting and, to my opinion, well presented and depicted in related figures.

The discussion is well written, summarizing and discussing the main findings of the review. I think that the existence of a paragraph summarizing the main limitations of this review adds to the scientific value of this paper.

Regarding “conclusions”, a section proposing some specific targets for future studies could be added.

References are adequate in number and relative to the subject.

English language and style are generally fine but there are some minor issues that need to be addressed before publication (for example some long sentences could be separated in shorter ones, to make the text more comprehensible).

Author Response

Dear Revisor,

We are very appreciative of the careful reading which you gave our manuscript. We meticulously addressed each of the points and questions raised and performed the corrections or justifications in accordance. In the discussion that follows we have indicated our direct responses to the questions raised by your thorough review. We believe the manuscript significantly improved because of your constructive criticisms and hope that you will find the revised version of this manuscript acceptable. We have enhanced the aforementioned paragraph to achieve a more comprehensive textual representation. Also, we have introduced a prospective direction within the concluding section. Furthermore, we have optimized the language by decomposing lengthy sentences, thus rendering the text more digestible and intelligible.

Reviewer 4 Report

Review of Scientific Article: "Effects of Caffeine on Symptoms in Children with ADHD: A Systematic Review and Meta-Analysis of Randomized Trials"

The scientific article titled "Effects of Caffeine on Symptoms in Children with ADHD: A Systematic Review and Meta-Analysis of Randomized Trials" presents a comprehensive study that examines the potential impact of caffeine on ADHD symptoms in children. The abstract provides a succinct overview of the paper, outlining the research objective, methodologies employed, and main findings. While the results may not be groundbreaking and the sample size of the analyzed articles is relatively small, the paper adheres to scientific standards and could contribute to the ongoing discourse surrounding ADHD therapy.

The introduction effectively contextualizes ADHD and its existing treatment methods, laying the foundation for the research question. Referencing animal studies that suggest a positive link between caffeine and ADHD symptoms adds depth to the exploration.

The methods section is well-detailed, describing source selection with a clear indication of careful research methodology. The utilization of reputable databases and the comparison of caffeine against placebo in randomized controlled trials is an appropriate approach to yield credible results.

The results section is presented in detail, providing a comprehensive overview of the collected data. The acknowledgment of the limited sample size is crucial, as it may impact the generalizability of the conclusions. The breakdown of results for each aspect of ADHD offers a nuanced understanding of the complexity of the issue.

The conclusion is thoughtfully formulated, grounded in a holistic interpretation of the gathered evidence. The article indicates that despite some positive outcomes in individual studies, the overall analysis does not strongly support significant benefits of caffeine in treating ADHD. This realistic approach underscores the need for further research in the realm of caffeine as a therapy for children with ADHD.

Author Response

Dear Revisor,

We are very appreciative of the careful reading which you gave our manuscript. We meticulously addressed each of the points and questions raised and performed the corrections or justifications in accordance. In the discussion that follows we have indicated our direct responses to the questions raised by your thorough review. We believe the manuscript significantly improved because of your constructive criticisms and hope that you will find the revised version of this manuscript acceptable.

Round 2

Reviewer 1 Report

Dear authors, 

Thank you for providing thorough responses to the comments.

The paper has shown remarkable improvement. However, prior to finalizing my acceptance of the revisions, I would like to highlight a few points:

Regarding question 4: 

Please include some information in the introduction on how caffeine improves dopamine, using this study on mice. Mentioning the regions of the brain that are under the influence such as nucleus accumbens. And then please briefly mention the role of nucleus accumbens in ADHD. https://www.ncbi.nlm.nih.gov/pmc/articles/PMC6758129/

Regarding question 10: 

The authors mentioned that gender has been added to table 1. I am seeing that the heading says: Patient (n)/ Sex (male). What is the reason that females are not mentioned? 

Regarding question 11: 

Please include point 11 in the paper and compare the duration of caffeine to that of ADHD medication, using valid evidence as you mentioned. 

Author Response

Review Report (Round 2):

Regarding question 4: Please include some information in the introduction on how caffeine improves dopamine, using this study on mice. Mentioning the regions of the brain that are under the influence such as nucleus accumbens. And then please briefly mention the role of nucleus accumbens in ADHD. https://www.ncbi.nlm.nih.gov/pmc/articles/PMC6758129/

Thank you for this brilliant suggestion. We now added this information to lines 135 to 139 of the introduction.

Regarding question 10: The authors mentioned that gender has been added to table 1. I am seeing that the heading says: Patient (n)/ Sex (male). What is the reason that females are not mentioned? 

Thank you for bringing this to our attention. There was no specific reason, so we have also made the modification in the manuscript by including the number of women in Table 1.

Regarding question 11: Please include point 11 in the paper and compare the duration of caffeine to that of ADHD medication, using valid evidence as you mentioned. 

Response: Thank you very much for your input. We have added this addendum to lines 347 to 350 of the manuscript.